# Persistent Activation of the P2X7 Receptor Underlies Chronic Inflammation and Carcinogenic Changes in the Intestine

**DOI:** 10.3390/ijms252010874

**Published:** 2024-10-10

**Authors:** Patricia Teixeira Santana, Isadora Schmukler de Lima, Karen Cristina da Silva e Souza, Pedro Henrique Sales Barbosa, Heitor Siffert Pereira de Souza

**Affiliations:** 1Department of Clinical Medicine, Federal University of Rio de Janeiro, Rio de Janeiro 21941-913, Brazil; pattsant@gmail.com (P.T.S.); isadora.schmukler@gmail.com (I.S.d.L.); karenecris480@gmail.com (K.C.d.S.e.S.); phsalesbarbosa@gmail.com (P.H.S.B.); 2D’Or Institute for Research and Education (IDOR), Rua Diniz Cordeiro 30, Botafogo, Rio de Janeiro 22281-100, Brazil

**Keywords:** P2X7, ATP, inflammatory bowel disease, colorectal cancer, damage-associated molecular patterns

## Abstract

Aberrant signaling through damage-associated molecular patterns (DAMPs) has been linked to several health disorders, attracting considerable research interest over the last decade. Adenosine triphosphate (ATP), a key extracellular DAMP, activates the purinergic receptor P2X7, which acts as a danger sensor in immune cells and is implicated in distinct biological functions, including cell death, production of pro-inflammatory cytokines, and defense against microorganisms. In addition to driving inflammation mediated by immune and non-immune cells, the persistent release of endogenous DAMPs, including ATP, has been shown to result in epigenetic modifications. In intestinal diseases such as inflammatory bowel disease (IBD) and colorectal cancer (CRC), consequent amplification of the inflammatory response and the resulting epigenetic reprogramming may impact the development of pathological changes associated with specific disease phenotypes. P2X7 is overexpressed in the gut mucosa of patients with IBD, whereas the P2X7 blockade prevents the development of chemically induced experimental colitis. Recent data suggest a role for P2X7 in determining gut microbiota composition. Regulatory mechanisms downstream of the P2X7 receptor, combined with signals from dysbiotic microbiota, trigger intracellular signaling pathways and inflammasomes, intensify inflammation, and foster colitis-associated CRC development. Preliminary studies targeting the ATP−P2X7 pathway have shown favorable therapeutic effects in human IBD and experimental colitis.

## 1. Introduction

Damage-associated molecular patterns (DAMPs) are endogenous stress molecules released during cell or tissue injury. DAMPs can arise from various cellular and tissue sources, including the cytosol, nucleus, mitochondria, and extracellular matrix [1]. These molecules can be released due to exogenous or endogenous stimuli and operate as signaling mediators of stress and the immune response through membrane or intracellular receptors or following endocytosis [2,3]. DAMPs are also regarded as pro-inflammatory mediators that can amplify the inflammatory response, fueling the development of pathological changes underlying several inflammatory and autoimmune disorders, such as psoriasis, rheumatoid arthritis, and multiple sclerosis [4,5,6,7,8,9].

The P2X7 receptor (P2X7R) is a trimeric transmembrane protein belonging to the adenosine triphosphate (ATP)-activated P2X receptor family. Upon exposure to low concentrations of extracellular ATP, the P2X7R undergoes a conformational change, leading to the opening of ion channels that are permeable to Na^+^, K^+^, and Ca^2+^ [10]. In contrast, when continuously stimulated with high concentrations of ATP, the P2X7R opens large pores that are permeable to macromolecules [11]. Upon activation of the ATP−P2X7R pathway, various potential biological actions, including those of cells, tissues, and microbiota, can be triggered according to the local environment. P2X7R activation has been implicated in many different physiological and pathological processes, such as chemoattraction, metabolism, autophagy, cell proliferation [12], invasion, cell death, cytokine production, inflammasome activation [13], and defense against microorganisms [14] (Figure 1). The gastrointestinal tract, which represents the largest surface of contact with the external environment, has the largest immune cell population and neural network in the body, and the immensely diverse ecosystem that constitutes the gut microbiota comprises a complex scenario in which P2X7 is expected to have an even more intricate participation both in normal homeostasis and in pathological conditions, from inflammation to cancer.

This review focuses on fundamental mechanisms and potential relationships between an abnormal ATP−P2X7 pathway function and the development and progression of inflammatory bowel disease (IBD) and colorectal cancer (CRC).

## 2. Rise of Inflammatory Bowel Disease and Colorectal Cancer

The global increase in chronic non-communicable diseases, including immune-mediated inflammatory diseases (IMIDs) and gastrointestinal cancers, has been associated with environmental and societal changes that have accelerated in the last century [15,16,17,18]. Concerning chronic noncommunicable diseases, ubiquitous evolutionary modifications in most societies typically encompass sociocultural adjustments, including industrialization, dietary changes, sanitation, urban population agglomeration, antibiotics, and vaccination [19]. Such progressive changes, referred to as Westernized lifestyles, also promote microbiome alterations and genetic and epigenetic modifications [20,21]. Several shared predisposing environmental factors and pathogenic mechanisms explain the link between chronic inflammatory diseases and gastrointestinal cancers [22,23].

IBD, comprising Crohn’s disease and ulcerative colitis, is a complex chronic inflammatory disorder of unknown cause with unmet therapeutic needs [24]. The increasing incidence and prevalence of IBD worldwide, including developing countries in Asia and South America, underscore the impact of recent local environmental and societal modifications [25]. Currently, IBD has been thought to develop when an individual with a predisposing genetic background is exposed to specific enteric infections, dietary elements, pollutants, and xenobiotics, among other factors [26]. Such an environment and lifestyle, regarded as Westernized, favors intestinal dysbiosis, in which more aggressive microbiota strains may prevail and determine the inflammatory profile [27].

CRC is the third most common cancer worldwide, with an incidence of 1.9 million new cases and 930,000 deaths in 2020. A projection for 2040 indicates an increase to 3.2 million new cases and 1.6 million deaths, highlighting its importance as a public health issue [28,29]. Therefore, a better understanding of the disease and search for biomarkers can result in a more efficient approach, favoring greater screening capacity, earlier diagnosis, and a relevant impact on disease outcomes. The worldwide rise of CRC, the most common gastrointestinal tumor, has been primarily attributed to lifestyle modifications associated with industrialization and economic development, accompanied by dietary modifications, pollution, sedentarism, smoking, diabetes mellitus, and obesity, among others [30,31,32].

## 3. Purinergic Signaling in Inflammatory Bowel Diseases and Colorectal Cancer

DAMPs are released owing to cell or tissue damage, are regarded as pro-inflammatory mediators, and are associated with acute and chronic inflammatory disorders, including IBD [1]. High concentrations of extracellular ATP, released into the extracellular medium, activate the P2X7R under most inflammatory conditions [33]. The P2X7R is activated by extracellular ATP, which acts as a DAMP after cell damage during inflammation, leading to the activation of extracellular signal-regulated protein kinases 1 and 2 (ERK1/2), and the NLRP3 inflammasome, culminating in the production of pro-inflammatory mediators and cytokines such as IL-1b [11]. In the gut, inflammation and dysbiosis cause loss of membrane integrity and cell death, leading to the release of ATP, which in high extracellular concentrations activates the P2X7R [34,35].

Our group previously demonstrated that the activation of the P2X7R by ATP induces apoptosis and autophagy in human epithelial cells, possibly via reactive oxygen species (ROS) production [36]. Autophagy is a sign of metabolic burden in the cell during the inflammatory response and under stress conditions, and P2X7R-induced ROS production can lead to its activation [37]. The stimulation of the P2X7R triggers ROS production, and both processes lead to NLRP3 inflammasome activation, elevate inflammatory cytokine levels, and promote inflammation [36,38]. In mice experimental models, we showed that the P2X7R plays a crucial role during inflammation by triggering death and retention of regulatory T cells in the mesenteric lymph nodes [39] and that the prophylactic systemic P2X7R blockade effectively prevents experimental colitis [40]. Moreover, P2X7R-deficient animals did not develop chemically induced colitis. In addition, the overexpression of the P2X7R in the inflamed mucosa of patients with IBD is consistent with the involvement of purinoceptors in cell death and inflammation [41]. In addition, the chronic inflammatory process underlying IBD may give rise to mutations and establish a regenerative environment that determines the selective pressure favoring the development of colitis-associated colorectal cancer (CA-CRC) [42]. In a CA-CRC experimental model, the ATP/P2X7R pathway was implicated in inflammasome activation and complete development of colitis and colon tumors [43]. Current mechanistic and conceptual understanding supports the idea that purinergic signaling is involved in the pathogenesis of intestinal inflammation and tumorigenesis (Figure 2).

The P2X7R has been identified as a possible independent prognostic biomarker of CRC. Immunohistochemical analysis of samples from patients with CRC indicated high expression of the P2X7R in tumoral tissues compared to peritumoral sites [44,45]. High P2X7R expression was correlated with higher TNM stages (III and IV), long-distance metastasis, and lymph node invasion, mainly observed in these patients. Other poor prognoses, such as high-grade tumor dysplasia and high serum carcinoembryonic antigen (CEA) levels, have also been observed in patients with a high expression of this receptor. In all studies, high P2X7R expression was associated with low overall survival and poor prognosis, evidencing its potential as a prognostic biomarker [44,45,46]. More recent studies have correlated the loss-of-function P2X7R polymorphism E496A with a minor 5-year disease-free survival (DFS) in patients with STAGE III colon cancer who underwent adjuvant chemotherapy.

In vivo experiments showed that P2X7R-deficient mice injected with CT26 (murine colon carcinoma cells) had larger tumors than P2X7R-wild-type mice [47]. This suggests that the P2X7R may have different roles in tumor burden and the peritumoral tissue. In addition, P2X7R^-/-^ mice are more susceptible to tumor development in a colitis-associated protocol with azoxymethane and dextran sodium sulfate (AOM/DSS), resulting in a higher number of larger and more aggressive tumors [48]. Other in vivo analyses showed that P2X7R-overexpressing tumors present with more Ki67^+^ cells and a lower expression of cleaved caspase-3 [49].

In vitro experiments with P2X7R-overexpressed CT26, HCT-116, and RKO cells demonstrated an increase in the migration rate and ability to invade, as demonstrated by wound healing and transwell assays, respectively. Moreover, P2X7R overexpressed cells reduced E-cadherin levels and increased vimentin and N-cadherin expression, indicating an epithelial-to-mesenchymal transition (EMT) process and a prometastatic phenotype. In contrast, P2X7R knockdown reduced CT26 migration and invasion abilities and increased E-cadherin expression [50]. In another study, treatment of SW480 cells with ATP and BzATP significantly promoted vimentin, sail, and fibronectin protein and mRNA expression while reducing E-cadherin protein and mRNA expression. The competitive inhibition of the P2X7R by A438079 and AZD9056 reduced proliferation, migration, invasion, colony formation, and glycogen content levels in SW620 and HCT-116 cells and decreased ATP and BzATP induction [51,52]. In addition, P2X7R inhibition can reduce Bcl-2 and induce Bax and cleave caspases 3 and 9, suggesting a pro-apoptotic effect [51]. Therefore, as inflammation dually affects cancer development depending on its stage [53], the P2X7R may contribute differently to tumor initiation, promotion, and progression.

Moreover, the PANX1−P2X7R axis appears to play an essential role in the response to therapy. Treatment of CRC cell lines with TNFα and chemotherapeutic drugs (oxaliplatin and 5-fluorouracil) promoted PANX1 cleavage and the release of ATP to the extracellular space. Coculture with THP1 and immature dendritic cells (DC) promoted DC maturation markers (CD86, HLA-A, B2M, and CD80) and the production of pro-inflammatory cytokines (IL-6, IL-1β, and IL-18) after combined treatment, while knockdown of PANX1 reversed this effect. In vivo results showed that the knockdown of PANX1 in tumor cells reduced their response to oxaliplatin, thereby reducing anti-tumor immunity and tumor-infiltrating cells [54].

Although P2X7 receptor activation plays a role in cancer development, its dual functions are not fully understood, potentially due to differential expression profiles in immune and non-immune cells. Recent studies highlight the duality of the receptor in prostate cancer, demonstrating its prometastatic and proliferative effects through mechanisms involving the exosome/microvesicle, miRNA, and hypoxia-inducible factor-1 alpha (HIF-1α), which promote invasiveness and angiogenesis by elevating vascular endothelial growth factor (VEGF) levels [55,56]. Conversely, reducing P2X7 expression decreases HIF-1α and VEGF levels, thereby inhibiting angiogenesis, while P2X7 activation of ERK1/2 may facilitate cell apoptosis [57,58].

## 4. Abnormal Innate Immunity in Inflammatory Bowel Diseases and Colorectal Cancer

Innate immunity is crucial for initiating and directing adaptive immunity and seamlessly integrating cancer and immunity. When tumors are detected, innate immune cells are activated, enhancing their ability to execute certain functions and destroy the cancer cells. The destruction of tumor cells generates additional signals that further amplify the immune response [59]. Specific subsets of cells, particularly cytotoxic T cells, respond to transformed cells in a tumorigenic milieu. Innate immune cells, such as dendritic cells and macrophages, also sense that tumor cells are modified and activated, initiating an inflammatory response. Natural killer (NK) cells and antibody responses, which are highly oxidative, are critical in this scenario. Activation of the innate immune system produces cytokines and inflammatory mediators and shapes the adaptive response and efficiency [60].

However, depending on the site and inflammatory mediators, the innate immune response may promote tumor growth. In well-established tumors, where the inflammatory response is suppressed, the same cells from the innate immune system contribute to the maintenance of the tumor niche, angiogenesis, and metastasis by producing anti-inflammatory cytokines and matrix remodeling [60]. In the tumor microenvironment, cytokines and inflammatory mediators play a decisive role and direct the responses, promoting tumor cell death and clearance on one hand, and angiogenesis, metastasis, and other cellular processes involved in the onset of the disease on the other. Secreted cytokines can induce phenotypic changes and polarize immune cells [61]. Chronic intestinal inflammation is a well-known risk factor for CRC, and several animal models of CRC depend on intestinal inflammation for tumor development. Despite their different pathways and mechanisms, sporadic and colitis-associated tumors depend on cytokines for progression [62].

Oxidative stress is closely linked to inflammatory responses and has been implicated in gastrointestinal inflammatory diseases, including IBD, a chronic condition that primarily affects the small and large bowels [63]. These conditions arise from immune system overreaction and can lead to neoplastic transformation in some patients [64]. Research indicates that oxidative stress plays a significant role in the development of CRC [65,66]. Multiple factors influence CRC; however, the exact cause remains unclear. In addition to the impact of lifestyle factors, such as diet, smoking, stress, alcohol, and toxins, oxidative stress also plays a significant role in chronic inflammation and cancer development, causing epigenetic changes, affecting genetic predisposition, altering immune responses, and contributing to dysbiosis, which is an imbalance of the intestinal microbiota that is integral to CRC development [67] (Figure 1).

P2X7R activation leads to ROS production, and when stimulated at low levels, the P2X7R accelerates tumor development by boosting oxidative phosphorylation and aerobic glycolysis, which increases cellular ATP levels and provides a growth benefit [68,69]. P2X7R activation also triggers key growth pathways and stimulates the release of factors like VEGF and matrix metalloproteinases, which contribute to tumor progression, angiogenesis, and metastasis [58,69,70].

Macrophages are critical elements of the tumor and essential for regulating the innate immune system, maintaining tissue balance, and managing inflammation. These cells are drawn to tumors, where they differentiate into tumor-associated macrophages (TAMs), which can develop into two main types: M1 and M2. M1 macrophages, also referred to as pro-inflammatory macrophages, produce cytokines like IL-1β and TNF-α and are activated by lipopolysaccharides (LPS) and interferon-γ (IFN-γ). On the other hand, M2 macrophages are anti-inflammatory, secreting factors such as IL-10, TGF-β, and arginase 1 (Arg1), and are stimulated by IL-4 and IL-13. Initially, macrophages adopt an M1-like phenotype that helps destroy tumors. However, as cancer progresses, they often transition to an M2-like phenotype that facilitates tumor growth, with well-differentiated TAMs linked to poorer outcomes and lower survival rates [71,72].

In the tumor microenvironment (TME), macrophages interact with cancer cells, influencing tumor growth and spread. The P2X7R is highly expressed on macrophages and promotes an M2-immunosuppressive phenotype that hampers the immune attack on tumor cells [73,74]. Moreover, its absence reduces tumor growth and boosts T-cell activity [75]. P2X7 activation by eATP also enhances anti-tumor effects through cGAMP transfer and STING activation, and the eATP–P2X7–inflammasome–IL18 axis reduces macrophage numbers in tumors while improving T-cell responses [76,77]. Tumors in P2X7-knockout mice grow faster and have fewer macrophages than those in wild-type mice, indicating that targeting P2X7 on macrophages can enhance anti-tumor immunity [47].

Neutrophils, the most prevalent white blood cells in the bloodstream, transform into tumor-associated neutrophils (TANs) when migrating to tumors. TGF-β in the tumor environment drives their differentiation into either an N1 or N2 type. Neutrophils can both inhibit and promote cancer, and their depletion leads to increased tumor growth, proliferation, and aggressiveness [78,79]. P2X7R activation weakens their capacity to attack tumors, contributing to accelerated growth and metastasis, as observed in P2X7-knockout mice with diminished neutrophil presence [47,73,74].

NK cells, which originate from the bone marrow, are essential for combating tumors and modulating immune responses through their activating or inhibitory receptors. They kill cancer cells via the natural killer and antibody-dependent cellular cytotoxicity (ADCC) pathways. P2X7 receptor activation can hinder NK cell activity, reducing their ability to eliminate cancer cells and facilitating tumor expansion and spread, as tumors in P2X7-knockout mice show increased growth and decreased NK cell infiltration [47,73,74].

Dendritic cells (DCs) are specialized antigen-presenting cells that bridge the adaptive and innate immune systems and are vital for initiating and sustaining T-cell-mediated anti-tumor responses. A higher number of DCs in tumors is typically associated with a better prognosis [80]. In the TME, DCs use P2X7 receptor activation to trigger the NLRP3 inflammasome and IL-1β release, enhancing CD8^+^ and CD4^+^ T-cell responses, particularly after radiation and chemotherapy [81,82,83].

Myeloid-derived suppressor cells (MDSCs) are known for their immunosuppressive properties and the ability to regulate immune responses in various diseases. MDSCs, including granulocyte/polymorphonuclear cell MDSCs (PMN-MDSCs) and monocyte MDSCs (M-MDSCs), are associated with poor clinical outcomes because of their ability to suppress immune functions [84]. MDSCs are regulated by the P2X7 receptor, which affects their immunosuppressive role. P2X7 activation in MDSCs leads to the secretion of factors that inhibit the ability of the immune system to combat tumors, thereby facilitating tumor progression and metastasis [85,86].

Table 1 summarizes the multiple roles attributed to P2X7R modulation on intestinal immune cells in the context of IBD and CRC.

## 5. Inflammasome and DAMPS in Inflammatory Bowel Diseases and Colorectal Cancer

Innate receptors are essential for the innate immune response as they identify distinct molecules associated with microorganisms and cellular damage, such as pathogen- or microbial-associated molecular patterns (PAMPs or MAMPs) and DAMPs. These include nod-like receptors (NLRs), C-type lectin-like receptors (CLRs), RIG-I-like receptors (RLRs), and Toll-like receptors (TLRs). Recently, there has been a notable increase in the number of studies exploring the contribution of members of the NLR family to the pathobiology of IBD [92,93].

Cell death, such as necrosis, releases DAMPs, which are sensed and increase phagocytosis and clearance, amplifying inflammatory responses and eliciting antigen-specific adaptive immune responses. The activation profiles of these immune cells shape the adaptive immune response, and germline mutations in pattern recognition receptors (PRRs) can affect cancer risk and treatment [94]. Some of the DAMPS associated with tumor immunity are calreticulin, extracellular ATP, and high-mobility group box 1 protein (HMGB1), which favor DC maturation and priming of protective T-cell responses that can kill tumor cells and establish anti-tumor immunological memory [95].

DAMPs significantly influence immune interactions during intestinal inflammation and CRC progression. ATP, via the P2X7R, activates the NLRP3 inflammasome, attracts immune cells, and primes the adaptive immune system, affecting DCs to support an anti-tumor response and inhibit tumor growth [43,94]. HMGB1, which binds to TLR4, aids in tumor-antigen processing and presentation, enhancing the anti-tumor immune response through its effect on DCs [96]. TIM-3 reduces the immunogenicity of nucleic acids, affects DCs, and leads to a lower immune rejection of tumors [97]. Calreticulin facilitates tumor cell uptake by DCs and boosts chemotherapy-induced anti-tumor responses by reducing tumor growth after treatment [98]. S100A4 promotes angiogenesis and a pro-tumor immune environment, contributing to metastasis through the release of paracrine factors and pro-inflammatory cytokines, whereas S100A8/9, via the RAGE receptor, activates NF-κB and supports tumor cell growth [99,100]. IL-33, which binds to the IL-1R1 receptor, causes the accumulation of immunosuppressive cells, decreases natural anti-tumor immunity, increases cell proliferation, and stimulates angiogenesis, all of which contribute to tumor progression [101]. IL-1, through the IL-1R1 receptor, enhances cell activation and cytokine release, affecting endothelial cells, T cells, and macrophages, thus promoting tumor invasiveness and inflammation [102].

DAMPs have also been associated with inflammation and cancer. For example, autophagy defects can enhance intestinal inflammation [103]. Autophagy-deficient intestinal epithelial cells display increased apoptosis and generation of DAMPs, promoting inflammation [104]. Furthermore, in differentiated cells, such as those in the intestinal epithelium, autophagy may also result from oxidative and metabolic stimuli, consequently affecting repair, proliferation, and response to drug treatments [105].

The activation of PRRs leads to the assembly of inflammasomes, which are multiprotein complexes that sense and respond to extracellular and intracellular microbial components, thereby initiating inflammatory responses. Classically, their activation induces interleukin-1β (IL-1β) and IL-18 production and secretion in response to microbial signaling and endogenous molecules derived from host proteins [106]. Regarding such endogenous molecules, ATP emerges as a critical DAMP, as it activates the inflammasome, releasing IL-1β and IL-18 to induce inflammation and trigger pyroptosis through the interaction with the membrane P2X7R (Figure 1).

Key inflammasome components include NLR family members (e.g., NLRC4, NLRP1, and NLRP3), featuring a nucleotide-binding and oligomerization domain (NACHT) and either a caspase recruitment domain (CARD) or pyrin domain (PYD). The adaptor protein ASC, which includes the PYD and CARD domains, links NLRs to caspase-1 via PYD interactions [107]. These complexes are crucial for innate immune responses against microorganisms and autoinflammatory disorders [106,108]. This interaction causes ASC to aggregate and activate caspase-1, potentially accelerating tumor growth by increasing inflammatory cytokine production or inhibiting tumors by inducing apoptosis [109,110]. Activated caspase-1 then processes pro-inflammatory cytokines IL-1β and IL-18 and cleaves gasdermin D, leading to pyroptosis, a type of cell death involving the formation of pores in the cell membrane, which causes cell lysis and cytokine release [111,112]. Pyroptosis can be triggered through canonical (caspase-1) or non-canonical pathways (caspase-4/5 or caspase-11) [113]. Dysregulated inflammasome activity and excessive IL-1β and IL-18 production are associated with IBD and CRC. In turn, IBD increases CRC risk owing to factors such as disease duration and persistent mucosal inflammation, in parallel with the continuous stimulus for repair and regenerative mechanisms. Dysplasia, a precancerous change in the epithelial cells, is a significant factor in disease development among patients with IBD [114] (Figure 2).

The NLRP3 inflammasome has been extensively studied for its role in regulating tumors and shaping the tumor microenvironment. In mouse models of colon cancer induced by azoxymethane (AOM) and dextran sulfate sodium (DSS), NLRP3-knockout mice exhibited a higher likelihood of developing colon polyps and were more susceptible to AOM-DSS-induced colitis-associated colon cancer. These mice also showed accelerated tumor growth and reduced IL-18 levels in tissues. IL-18 is essential for epithelial barrier repair and has potential anti-tumor effects, as evidenced by the fact that IL-18-deficient mice are more prone to tumor development after AOM−DSS treatment [115]. In a similar mouse model, the absence of the P2X7 receptor, whose activation leads to NLRP3 inflammasome assembly, correlated with protection against CRC [43]. Furthermore, mice lacking ASC and caspase-1 are vulnerable to DSS-induced colitis and CRC, suggesting that NLRP3 inflammasome activation may inhibit colorectal carcinogenesis [116]. IL-18 secretion by the NLRP3 inflammasome can indirectly reduce tumor progression in colitis-associated colorectal cancer by inducing regulatory T (Treg) cells to produce IFN-γ and enhancing T-cell and NK-cell cytotoxicity. Thus, the NLRP3 inflammasome is crucial for an effective anti-tumor immune response [117].

Similarly, the NLRP1 inflammasome affects cancer progression. In inflammatory bowel disease and colorectal cancer, the NLRP1 inflammasome acts through IL-1β and IL-18. Reduced NLRP1 expression was found in colon cancer tissues compared to healthy controls, indicating its role in maintaining colon homeostasis [118]. In contrast, the NLRC4 inflammasome inhibits CRC by reducing cell proliferation and promoting cell death [119]. The AIM2 inflammasome, which recognizes DNA, suppresses AOM−DSS-induced colorectal cancer and spontaneous colorectal carcinogenesis by inhibiting colon cancer stem cell proliferation and promoting cell death [120]. In a model of non-alcoholic fatty liver disease (NAFLD), NLRC4 knockout resulted in increased growth and recurrence of liver metastases from colorectal cancer while significantly limiting hepatic tumor development. NLRC4-deficient mice with NAFLD also showed decreased tumor-associated macrophages (TAMs) and reduced IL-1β and VEGF expression, highlighting NLRC4’s role in managing the growth and recurrence of liver metastases in the context of both NAFLD and colorectal cancer [121].

## 6. P2X7R and Cell Fate in the Gastrointestinal Tract

In multicellular organisms, there is a constant effort to achieve a homeostatic balance between the number of new cells generated by mitosis and the number of damaged or spared cells removed from the body [122]. Cell death has long been considered a passive and degenerative process in cell injury, infection, and in the absence of growth factors [123]. However, not all processes that lead to cell death can be considered passive [124] because multicellular organisms can induce programmed cell death in response to intracellular or extracellular stimulation [125]. Cell death processes can be classified according to their morphological and biochemical characteristics into apoptosis, autophagy, necrosis, catastrophic mitosis, and senescence. Once cellular damage is detected, executioner caspases are activated, initiating a cascade of events that results in DNA fragmentation through the activation of endonucleases, destruction of nuclear and cytoskeletal proteins, cross-linking of proteins, expression of ligands for phagocytic cells, and the formation of apoptotic bodies [126]. The failure to regulate cell death can result in pathological changes and diseases. In the gut epithelium, which has one of the highest turnover rates in the body, the programmed cell death machinery is essential for homeostasis reestablishment after acute or chronic insults, preventing the dissemination of inflammatory processes [127]. Changes in the coordination of these types of cell death can result in excessive cell loss and barrier dysfunction, as observed in IBD [128], and uncontrolled growth and division of cells, as observed in tumorigenesis [129].

Numerous genetic studies have identified more than 200 polymorphisms associated with IBD. To date, most susceptibility genes are associated with defects in autophagy, endoplasmic reticulum stress, and microbial sensing, reinforcing the critical role of cell death in the pathogenesis of IBD [130,131]. High apoptosis rates have been detected in the intestinal epithelia of patients with IBD [132]. In CD, the percentage of apoptotic enterocytes is higher in inflamed areas than in non-inflamed areas and normal intestines, suggesting that apoptosis is directly associated with intestinal inflammation and increases in inflamed regions [133]. As expected, necrosis has been commonly identified in actively inflamed areas of the intestinal mucosa in patients with IBD [134]. Nevertheless, intermittent areas of necrosis were detected in patients with CD, even in the absence of acute inflammation, suggesting a possible predisposing defect [135]. Autophagy, a crucial mechanism for the cytoprotection and recycling of nonessential organelles under cellular stress conditions [136], is impaired in IBD [137]. In CD, identifying predisposing polymorphisms of the ATG16L1 gene by genome-wide association studies, which play a critical role in inflammation and microbial homeostasis, reinforces a pathogenic mechanism based on defective autophagy [138]. In addition to defects in Paneth cell functions [139], the ATG16L1 gene was also shown to regulate IL-1β and IL-18 production upon exposure to lipopolysaccharide [140]. However, autophagy may play a dual role in cancer development. Autophagy can be a tumor suppressor associated with apoptosis [141]. Conversely, abnormalities in autophagy can prompt cancer development when combined with defective apoptosis [142] and suppression of anti-tumor immunity, as observed in CRC [143].

Extracellular nucleotides, including ATP, play fundamental roles in many biological functions, including cell fate [144]. Previous studies have revealed that ATP treatment decreases cell growth in human intestinal epithelial cells [145], and autophagy is considered to control apoptosis under hypoxic conditions [146]. Furthermore, ATP plays a crucial role in the modulation of intestinal T-cell responses. The P2X7R is highly regulated in intestinal T cells, particularly in CD8^+^ T cells in the intestinal epithelium [147]. The P2X7R is expressed in the enteric ganglia of rodents, where it is localized in deep-seated primary afferent neurons, inhibitory motor neurons, and glial cells [148,149].

Notably, the P2X7R is both functional and responsive to ATP-induced apoptosis, thus emerging as a novel regulatory factor that influences T-cell responses in the intestinal mucosa [150]. When ATP is present at high concentrations, it induces P2X7R activation and cell death [151]. The prolonged activation of the P2X7R in macrophages by millimolar concentrations of ATP triggers cell death via apoptosis and necrosis [152]. However, Baricordi et al. have demonstrated that this receptor participates in lymphoid cell proliferation [153]. The P2X7R, which influences inflammation and dysmotility, has also been investigated in IBD and influences inflammation and dysmotility [154].

In human intestinal epithelial cells, P2X7R activation by ATP induces apoptosis and autophagy, probably via ROS production, which may have implications for gut inflammatory conditions [36]. Apoptosis prompted upon P2X7R stimulation is associated with increases in cytosolic Ca^2+^ and ROS, along with endoplasmic reticulum stress, which is also known to induce necroptosis [155]. Positive allosteric modulation of P2X7 promoted an intrinsic apoptosis pathway characterized by increased mitochondrial Ca^2+^, loss of mitochondrial membrane permeability, and exacerbated production of mitochondrial ROS [156]. In turn, the P2X7R blockade in a mouse model of colitis attenuated the inflammatory process that causes enteric cell death [148]. In addition, the P2X7R blockade in an in vivo colitis-associated cancer model contributed to Treg and neutrophil infiltration, epithelial cell growth, and reduced apoptosis, suggesting that the P2X7R may protect against cancer formation [11].

## 7. P2X7R and the Intestinal Microbiota

Microbes within the human body interact with the host at various locations, such as the skin and mucosal surfaces, which are critical for immune system homeostasis and cancer progression. In experimental models of IBD, the complete development of inflammatory changes depends on the presence of commensal bacteria that colonize the mucosa [157,158]. Approximately 13% of cancer cases worldwide are associated with chronic infections, and persistent inflammation plays a significant role in their development [159]. In addition to causing a clear inflammatory response, some well-described infections can also lead to cancer, including *Helicobacter pylori*, which predisposes patients to gastric cancer, hepatitis B or C virus infections leading to liver cancers, and human papillomavirus resulting in cervical cancer [160].

An imbalance in the composition and diversity of the microbial communities that colonize the body, known as dysbiosis, has been associated with several diseases. Long-term antibiotic use has been recognized as an independent risk factor for cancer [161]. Moreover, high-fat diets can disrupt the gut microbiome, potentially accelerating tumor growth in gastrointestinal cancers [162]. Antibiotic therapy reduces intestinal inflammation [163]. Moreover, probiotics effectively reduced clinical and endoscopic activity indices in patients with UC [164]. In germ-free animals, intestinal inflammation is much milder than in animals with preserved microbiota [165], highlighting the importance of microbiota in the full development of gut inflammation. In mouse models of liver cancer, bile acids produced by gut bacteria travel back to the liver, where they hinder the recruitment of NKT cells and reduce tumor control [166]. In experimental mouse models of CRC, damage to the intestinal lining allows microbial products to infiltrate early-stage tumors. Such infiltration triggers the activation of myeloid cells that produce IL-23, which boosts tumor-promoting cytokines such as IL-17 and IL-6 [167]. These insights highlight the crucial role of the gut microbiome in CRC and its broad implications for cancer development through systemic effects.

One of the basic mechanisms underlying the pathophysiology of IBD is a defect in epithelial barrier integrity, leading to a higher uptake of luminal microbial components and culminating in excessive immune cell activation [168]. In this scenario of epithelial barrier dysfunction and immune cell activation, ATP released from necrotic and apoptotic cells leads to activation of the ATP−P2X7R pathway [36], with the production of pro-inflammatory cytokines such as IL-1β and IL-18, which produce an inflammatory microenvironment also favorable to tumor development [43]. Other cytokines, such as IL-17A, TNF-α, and IL-6, activate NF-kappa β and signal transducer and activator of transcription 3 (STAT3), inducing tumor growth. The transcription factor NF-kappa β, per se, leads to the production of pro-IL-1β and pro-IL-18, perpetuating the pro-inflammatory environment and is involved with human tumorigenesis [169]. P2X7R activation is also involved, at least indirectly, in the production of IL-17A, TNF-α, and IL-6, as lower levels of these molecules were detected in explants’ supernatants of P2X7R^-/-^ mice in a model of CA-CRC [43].

In this context, dextran sodium sulfate, a well-established molecule used to induce colitis in mice through an increase in intestinal permeability, leads to a change in microbiota composition with an increase in the abundance of *Enterobacteriaceae*, *Bacteroidaceae*, and *Clostridium* spp. [170]. Using a CA-CRC model, our group elucidated decreased microbiota diversity in the AOM/DSS-induced P2X7R^+/+^ group [43]. Moreover, AOM/DSS-treated P2X7R^+/+^ mice showed a trend toward a higher relative abundance of Firmicutes, Tenericutes, and Fusobacterium phyla, with a highlight for *Mycoplasma* and *Mucispirillum* genera, compared to the control group. Similar to humans, the *Mycoplasma* genus is associated with CRC [171]. Therefore, the AOM/DSS murine model seems attractive for studying intestinal cancer and its microbiota association.

Regarding a purinergic influence on the microbiota composition, our group elucidated that P2X7^-/-^ mice had a higher relative abundance of Cyanobacteria and Spirochaetes phyla compared to their wild-type littermates [43], which could be one of the factors that may explain their protection against tumorigenesis. However, contrasting results were reported in another study [48], in which P2X7^-/-^ mice were more susceptible to tumorigenesis than wild-type mice. These data reinforce the importance of the gut microbiota in tumor development, as animal housing and care and microbiota composition differed between the two studies.

In a study analyzing fecal microbiota in a model of *Toxoplasma gondii*-induced ileitis, P2X7^-/-^ mice showed a trend toward decreased diversity in the ileitis-induced group [172]. This is consistent with the reduced microbial diversity observed in human patients with IBD [173]. In this study, Bacteroidetes and Firmicutes were the predominant phyla in all the studied groups, with no differences in their bacterial phyla. However, there was a difference in the relative microbial abundance between the ileitis-induced and control mice and between wild-type mice and their knockout littermates. P2X7^-/-^ mice with ileitis presented an increased relative abundance of the Cyanobacteria phylum, specifically the *Gastranaerophilales* order, in addition to an increase in the Bacteroidales order (Bacteroidetes phylum) and *Lachnospiraceae* family (Firmicutes phylum), compared to knockout uninfected mice. In wild-type mice, ileitis increased the relative abundance of the *Rhodospirillales* order (Proteobacteria phylum), the *Gastranaerophilales* order, and the *Bacteroidaceae* family compared to uninfected mice. Interestingly, the *Desulfovibrionaceae* family was reduced in the wild-type ileitis group compared with the control group, which is in disagreement with previous data that strongly associated increased levels of sulfate-reducing bacteria with IBD pathogenesis [173].

Patients with IBD have higher sulfate-reducing bacteria (SRB) levels in the colon and feces than healthy individuals [164,174]. However, whether such a profile represents a cause or a consequence is yet to be determined. Hydrogen sulfide, an SRB metabolic product, is also elevated in the feces of IBD patients [175]. In a previous study by our group, we demonstrated that *Desulfovibrio indonesians* and a human SRB consortium could induce colitis in germ-free mice and aggravate colon inflammation in TNBS-treated animals [176]. These data corroborate the essential role of SRB in the disease pathogenesis and perhaps a potential therapeutic target.

Hence, our group confirmed that purinergic receptors have an essential function in DAMP recognition and response formulation. As discussed above, this response may assume a more pro-inflammatory or anti-inflammatory profile depending on the microenvironment, composition, and diversity of the local microbiota [173]. Similarly, this response may control and regulate the microbiota population, as discussed previously [36,43]. Thus, the process of tumorigenesis can be influenced by purinergic activity, as the purinergic pathway leads to the activation of transcriptional factors involved with tumor growth (NF-kappa β and STAT3). Moreover, the activation of purinergic pathways culminates with the production of pro-inflammatory cytokines, such as IL-1β and IL-18, which contribute to an inflammatory microenvironment predisposing to tumor development (Figure 1). Therefore, the development of intestinal tumorigenesis also depends on the presence of signals from the dysbiotic microbiota and is partially controlled by the action of the ATP-P2X7R pathway, which modulates the composition and diversity of these microorganisms during chronic inflammation [36,43,172].

## 8. Conclusions

In the gastrointestinal tract, P2X7R is present in both immune and non-immune cells and has pleiotropic effects. Owing to the complexity of the intestinal structure and function, with many different cell types and site-specific activities combined with signals from the body’s largest microbiota, the P2X7R responds according to the local context and dynamic changes that might ensue with time. The ATP−P2X7R pathway is involved in cell death, autophagy, release of pro-inflammatory cytokines, inflammasome activation, chemoattraction, and defense against microorganisms, in addition to cell invasion, proliferation, and carcinogenesis. The broad range of biological functions of the epithelium and lamina propria position the P2X7R as a critical regulator of intestinal homeostasis. However, in extreme conditions, such as chronic inflammatory disorders like IBD, the ATP−P2X7R pathway amplifies the inflammatory response involving immune and non-immune cells, fostering the development of pathological changes related to particular disease phenotypes. In chronic inflammation, persistent exposure to various inflammatory mediators and oxidative stress, along with signals from the dysbiotic microbiota, results in the release of additional DAMPs, including ATP, and the overexpression of the P2X7R. These chronic changes related to the ATP−P2X7R pathway drive epigenetic reprogramming in IBD and underlie modifications that favor intestinal carcinogenesis. Together, these observations implicate excessive purinergic signaling in the pathogenesis of chronic intestinal inflammation and carcinogenesis. Therefore, the P2X7R emerges as a potential novel target for treating IBD and lowering the risk of CRC, particularly in neoplastic changes resulting from chronic inflammation.

## Figures and Tables

**Figure 1 ijms-25-10874-f001:**
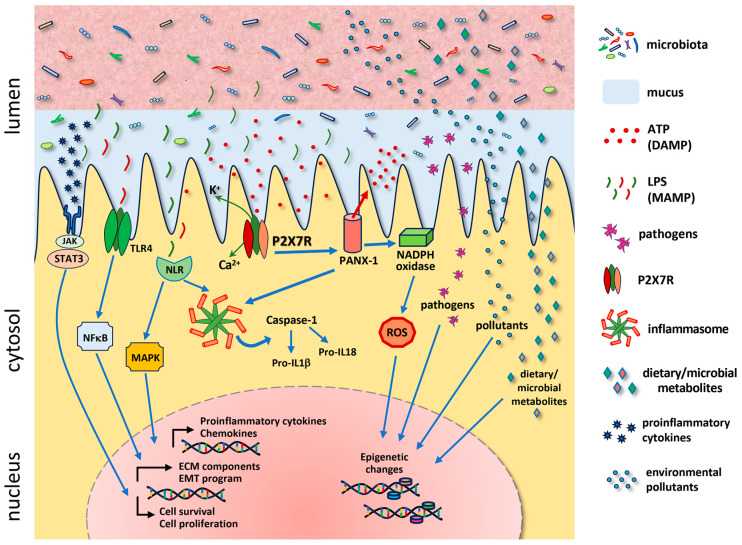
Schematic representation of a cell expressing the P2X7 receptor (P2X7R) in the intestinal mucosa. In the extracellular milieu, adenosine triphosphate (ATP) functions as a damage-associated molecular pattern (DAMP) signaling molecule that activates the P2X7R. Upon stimulation with low concentrations of ATP, the P2X7R promotes channel opening for Na^+^, K^+^, and Ca^2+^. In contrast, high concentrations or long-term stimulation with ATP leads to the opening of a non-selective pore that allows the passage of macromolecules. P2X7R stimulation activates Pannexin-1, a membrane channel that mediates the production of reactive oxygen species (ROS), and activates the inflammasome, driving the inflammatory response. In parallel, the nucleotide-binding-oligomerization-domain (NOD)-like receptor (NLR) detects microbial signals in the cytosol and activates the mitogen-activated protein kinase (MAPK) pathway and the inflammasome. Microbe-associated molecular patterns (MAMPs) like lipopolysaccharide (LPS) are also detected through toll-like receptors on the cell surface and prompt inflammatory responses through the nuclear factor kappa B (NFκB) pathway. A low grade of cytokines released following inflammation may trigger a para-inflammatory reaction, activating signal transducer and activator of transcription 3 (STAT3) through the Janus kinase (JAK) pathway. Depending on the cell type involved and the timing of the pathologic process, the JAK−STAT and NFκB pathways may promote the production of extracellular matrix (ECM) components underlying fibrogenesis, activate the epithelial-to-mesenchymal transition (EMT) program, and regulate the cell survival and proliferation pathways underlying both tissue repair and carcinogenesis. Additional factors related to environmental exposure, including microbial pathogens, dietary components, and environmental pollutants, combine with ROS to mediate epigenetic reprogramming.

**Figure 2 ijms-25-10874-f002:**
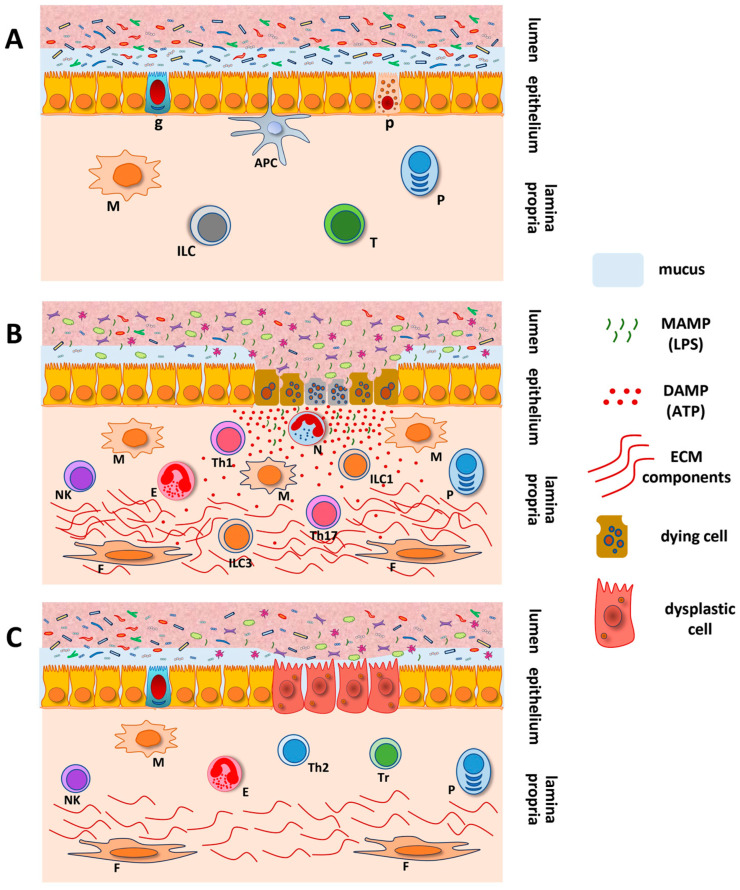
Schematic representation of inflammation and neoplastic transformation in the gut. Under normal circumstances, homeostatic mechanisms preserve the architecture and functionalities of the intestinal mucosa. In normal conditions, the lamina propria comprises a diverse immune cell population in a predominantly resting state (**A**). Under chronic inflammatory conditions, immune cells increase dramatically in the lamina propria. In addition to gut microbiota-derived microbe-associated molecular patterns (MAMPs), damage-associated molecular patterns (DAMPs) such as ATP that are released due to tissue damage and cell death activate Th1/Th17-effector cells and the inflammasome, further compromising the epithelial barrier. In prolonged dysbiosis, a secondary inflammatory response mediated by immune and non-immune cells amplifies the inflammatory response and induces wound healing mechanisms, resulting in tissue remodeling, including the overproduction of extracellular matrix (ECM) components (**B**). Aberrations in the highly organized process of re-epithelialization and tissue remodeling in response to chronic inflammation and signals from a dysbiotic microbiota may result in cellular reprogramming associated with malignant transformation (**C**). APC, antigen-presenting cell; g, goblet cell; p, Paneth cell; P, plasma cell; M, macrophage; ILC, innate lymphoid cell; T, T-cell; N, neutrophil; E, eosinophil; NK, natural killer cell; F, fibroblast; Th1, Th2, and Th17, effector T-helper cell types; Tr, T regulatory cell; ILC1, ILC3, innate lymphoid cell type 1 and type 3.

**Table 1 ijms-25-10874-t001:** Role of the P2X7R on intestinal immune cells in the context of inflammation and cancer.

Immune Cell	P2X7R Activation	Effect on IBD	Effect on CRC	References
Macrophages	Leads to the activation of the NLRP3 inflammasome and the ERK1/2, NFκB, and PI3K/AKT pathways	The M1 phenotype is pro-inflammatory, producing IL-1β, IL-18, TNF-α and induction of pyroptotic cell death, contributing to chronic inflammation and tissue damage	The M2 phenotype is immunosuppressive and facilitates tumor growth, angiogenesis, and metastasis by secreting anti-inflammatory cytokines (IL-10 and TGF-β)	[71,72,73,75,76]
T-cells	Plays a role in the homeostatic regulation of subpopulations by inducing P2X7R-dependent T-cell death; favors differentiation, migration, and induces the production of MMPs; and induces NFAT and MAPK signaling, promoting IL-2 expression and T-cell proliferation	Increased activated CD4^+^ T-cells orchestrates Th1/Th17 immune responses; survival and differentiation of CD8^+^ cytotoxic T-cells that promote tissue damage by releasing perforin and granzymes; inhibition of the suppressive potential of Treg	Reduction of Tregs and of CD8^+^ T-cell infiltration enhance anti-tumor responses; induction of cellular senescence in tumor-infiltrating T-cells	[87,88,89,90,91]
Neutrophils	Triggers the NLRP3 inflammasome and IL-1β release	Production of excessive proinflammatory cytokines, MMPs, and ROS, fueling inflammation and tissue damage	Reduced tumor-fighting capacity; differentiate into TANs in the tumor microenvironment; the N2 type promotes tumor growth and metastasis	[47,73,74] [78,79]
Dendritic Cells (DCs)	Enhances antigen presentation, amplifying the responses of CD4^+^ T cells and CD8^+^ T cells; triggers the NLRP3 inflammasome and IL-1β release in the context of the TME	Overactive DCs produce excessive pro-inflammatory cytokines, leading to chronic inflammation	Dysfunctional DCs fail to present antigens effectively, inducing tumor growth and immune evasion	[80,81,82,83]
Natural Killer (NK) Cells	Reduces the ability of NK cells to kill cancer cells	Reduced activity can lead to less effective immune surveillance and increased inflammation	Impaired NK cell function facilitates tumor expansion and metastasis	[47,73,74]
Myeloid-Derived Suppressor Cells	Leads to the production of immunosuppressive factors, including ROS, Arginase-1, and TGFβ	Suppression of the immune responses, exacerbating chronic inflammation that can worsen the tissue damage in the gut and IBD symptoms	Increases VEGF, stimulating angiogenesis and blood supply with nutrients and facilitating metastasis by inhibiting anti-tumor immunity	[84,85,86]

IBD, inflammatory bowel disease; CRC, colorectal cancer; NFAT, nuclear factor of activated T cells; MAPK, mitogen-activated protein kinases; TME, tumor microenvironment; ROS, reactive oxygen species; MMPs, matrix metalloproteinases; TAN, tumor-associated neutrophils; VEGF, vascular endothelial growth factor.

## Data Availability

Not applicable.

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
