# Peer review of "Persistent Activation of the P2X7 Receptor Underlies Chronic Inflammation and Carcinogenic Changes in the Intestine"

_ijms, 2024, doi:10.3390/ijms252010874_

Round 1

Reviewer 1 Report

Comments and Suggestions for Authors

Scholarly Review. It is comprehensive, and therefore dense to read, but I think the authors have done a god job communicating the complexities and issues. Perhaps the authors could highlight more the bimodal paradoxical effects of over and under expression of the P2X7 receptor and explain the interpetation of this more clearly

There is no refeerence to the volume of work in the prostate available for this cytokine and receptor; there should be.

Author Response

REVIEWER 1

It is comprehensive, and therefore dense to read, but I think the authors have done a god job communicating the complexities and issues. Perhaps the authors could highlight more the bimodal paradoxical effects of over and under expression of the P2X7 receptor and explain the interpretation of this more clearly

There is no reference to the volume of work in the prostate available for this cytokine and receptor; there should be.

R: We thank this reviewer for supporting our study and for his/her suggestions. We included new sentences regarding the requested subject and four additional references.

Although P2X7 receptor activation plays a role in cancer development, its dual functions are not fully understood, potentially due to differential expression profiles in immune and intestinal cells. Recent studies highlight the duality of the receptor in prostate cancer, demonstrating its prometastatic and proliferative effects through mechanisms involving exosome/microvesicle, miRNA, and hypoxia-inducible factor-1 alpha (HIF-1α), which promote invasiveness and angiogenesis by elevating vascular endothelial growth factor (VEGF) levels (Qiu Y. et al., 2014; Pegoraro A., et al. 2021). Conversely, reducing P2X7 expression decreases HIF-1α and VEGF levels, thereby inhibiting angiogenesis, while P2X7 activation of ERK1/2 may facilitate cell apoptosis (Tafani M. et al., 2011; Wang Z. et al., 2023).

References

  1. Qiu, Y.; Li, W.H.; Zhang, H.Q.; Liu, Y.; Tian, X.X.; Fang, W.G. P2X7 mediates ATP-driven invasiveness in prostate cancer cells. PLoS One 2014, 9, e114371, doi:10.1371/journal.pone.0114371.
  2. Pegoraro, A.; De Marchi, E.; Ferracin, M.; Orioli, E.; Zanoni, M.; Bassi, C.; Tesei, A.; Capece, M.; Dika, E.; Negrini, M., et al. P2X7 promotes metastatic spreading and triggers release of miRNA-containing exosomes and microvesicles from melanoma cells. Cell Death Dis 2021, 12, 1088, doi:10.1038/s41419-021-04378-0.
  3. Wang, Z.; Zhu, S.; Tan, S.; Zeng, Y.; Zeng, H. The P2 purinoceptors in prostate cancer. Purinergic Signal 2023, 19, 255-263, doi:10.1007/s11302-022-09874-2.
  4. Tafani, M.; Schito, L.; Pellegrini, L.; Villanova, L.; Marfe, G.; Anwar, T.; Rosa, R.; Indelicato, M.; Fini, M.; Pucci, B., et al. Hypoxia-increased RAGE and P2X7R expression regulates tumor cell invasion through phosphorylation of Erk1/2 and Akt and nuclear translocation of NF-kappaB. Carcinogenesis 2011, 32, 1167-1175, doi:10.1093/carcin/bgr101.

Reviewer 2 Report

Comments and Suggestions for Authors

The study titled “Persistent activation of the P2X7 receptor underlies chronic inflammation and carcinogenic changes in the intestine” provides a comprehensive overview of the role of DAMPs-ATP, in activating the purinergic receptor P2X7 and its implications in IBD and CRC. However, this study should further discuss the therapeutic potential of targeting the P2X7 pathway and include how P2X7 interacts with dysbiosis microbiota to drive disease progression. Additionally, I have suggestions and questions to enhance this manuscript.

1. The studies discussing the role of P2X7R in regulating macrophages and tumor-associated neutrophils are compelling.  However, providing more specific references to experimental findings, particularly from in vivo models like P2X7R KO mice, would offer better insight into how the absence of P2X7R influences tumor progression and immune response.

2. In section 3, how ROS and autophagy specifically contribute to these processes in the context of P2X7R activation?

3. In sections 3 and 5, the role of DAMPs as pro-inflammatory mediators in inflammatory disorders, including IBD, is clear. However, the mechanisms underlying the interaction between DAMPs and P2X7R remain unclear. What is the mechanism through which DAMPs activate P2X7R, and how does this interaction contribute to the inflammatory response in IBD?

4. In section 6, the mechanism by which P2X7R activation leads to both apoptosis and autophagy, via ROS reaction/production, needs to be clarified. Additional details and further discussion are required on the downstream pathways linking P2X7R activation to ROS production.

Author Response

REVIEWER 2

The study titled “Persistent activation of the P2X7 receptor underlies chronic inflammation and carcinogenic changes in the intestine” provides a comprehensive overview of the role of DAMPs-ATP, in activating the purinergic receptor P2X7 and its implications in IBD and CRC. However, this study should further discuss the therapeutic potential of targeting the P2X7 pathway and include how P2X7 interacts with dysbiosis microbiota to drive disease progression.

Additionally, I have suggestions and questions to enhance this manuscript.

  1. The studies discussing the role of P2X7R in regulating macrophages and tumor-associated neutrophils are compelling.  However, providing more specific references to experimental findings, particularly from in vivo models like P2X7R KO mice, would offer better insight into how the absence of P2X7R influences tumor progression and immune response.

R: We understand this reviewer’s point of view and amended the manuscript to incorporate his/her suggestions.

  1. In section 3, how ROS and autophagy specifically contribute to these processes in the context of P2X7R activation?

R: We amended the text and included new references to support the explanations regarding this specific subject.

Autophagy is a sign of metabolic burden in the cell during the inflammatory response and stress conditions, and P2X7R-induced ROS production can activate it [Zhang et al., 2022]. The stimulation of P2X7R triggers ROS production, and both processes lead to NLRP3 inflammasome activation, elevate inflammatory cytokine levels, and promote inflammation [34; Kong et al., 2022]. 

  1. Zhang, R.; Kang, R.; Klionsky, D.J.; Tang, D. Ion Channels and Transporters in Autophagy. Autophagy 2022, 18, 4-23, doi:10.1080/15548627.2021.1885147.
  2. Kong, H.; Zhao, H.; Chen, T.; Song, Y.; Cui, Y. Targeted P2X7/NLRP3 signaling pathway against inflammation, apoptosis, and pyroptosis of retinal endothelial cells in diabetic retinopathy. Cell Death Dis 2022, 13, 336, doi:10.1038/s41419-022-04786-w.

  1. In sections 3 and 5, the role of DAMPs as pro-inflammatory mediators in inflammatory disorders, including IBD, is clear. However, the mechanisms underlying the interaction between DAMPs and P2X7R remain unclear. What is the mechanism through which DAMPs activate P2X7R, and how does this interaction contribute to the inflammatory response in IBD?

R: We understand this reviewer’s concern and have included additional explanatory text on the ATP-P2X7R mechanistic role in inflammation, as well as new references.

The P2X7R is activated by extracellular ATP, which acts as a DAMP after cell damage during inflammation, leading to the activation of extracellular signal-regulated protein kinases 1 and 2 (ERK1/2) and NLRP3 inflammasome, culminating in the production of pro-inflammatory mediators and cytokines such as IL-1b [11]. In the gut, inflammation and dysbiosis cause loss of membrane integrity and cell death, leading to the release of ATP that, in high extracellular concentrations, activates the P2X7R [34,35].

  1. Cheng N, Zhang L, Liu L. Understanding the Role of Purinergic P2X7 Receptors in the Gastrointestinal System: A Systematic Review. Front Pharmacol. 2021 Dec 20;12:786579. doi: 10.3389/fphar.2021.786579. PMID: 34987401; PMCID: PMC8721002.
  2. Burnstock, G.; Jacobson, K.A.; Christofi, F.L. Purinergic drug targets for gastrointestinal disorders. Curr Opin Pharmacol 2017, 37, 131-141, doi:10.1016/j.coph.2017.10.011.

  1. In section 6, the mechanism by which P2X7R activation leads to both apoptosis and autophagy, via ROS reaction/production, needs to be clarified. Additional details and further discussion are required on the downstream pathways linking P2X7R activation to ROS production.

R: We added sentences to Chapter 6, including two new references, as follows:

Apoptosis prompted upon P2X7R stimulation is associated with increases in cytosolic Ca++ and ROS along with endoplasmic reticulum stress, also known to induce necroptosis [150]. Positive allosteric modulation of P2X7 was shown to promote an intrinsic apoptosis pathway, characterized by an increase in mitochondrial Ca2+, loss of mitochondrial membrane permeability, and exacerbated production of mitochondrial ROS [151].

  1. Vandenabeele, P.; Galluzzi, L.; Vanden Berghe, T.; Kroemer, G. Molecular mechanisms of necroptosis: an ordered cellular explosion. Nat Rev Mol Cell Biol 2010, 11, 700-714, doi:10.1038/nrm2970.
  2. Bidula, S.; Dhuna, K.; Helliwell, R.; Stokes, L. Positive allosteric modulation of P2X7 promotes apoptotic cell death over lytic cell death responses in macrophages. Cell Death Dis 2019, 10, 882, doi:10.1038/s41419-019-2110-3.

Reviewer 3 Report

Comments and Suggestions for Authors

The study demonstrates that purinergic signaling is involved in the pathogenesis of chronic intestinal inflammation and carcinogenesis and the role of P2X7R as a potential novel target for treating inflammatory bowel disease and lowering the risk of colitis-associated colorectal cancer.

8. Conclusion may be revised to focus on the role of P2X7R and describe how P2X7R lowers the risk of colitis-associated colorectal cancer.

The abbreviations such as P2X7R, CRC or CA-CRC need to be consistent in the manuscript.

Please re-check the manuscript, especially Ca2+ in lines 44 and 61, etc.

Author Response

REVIEWER 3

The study demonstrates that purinergic signaling is involved in the pathogenesis of chronic intestinal inflammation and carcinogenesis and the role of P2X7R as a potential novel target for treating inflammatory bowel disease and lowering the risk of colitis-associated colorectal cancer.

  1. Conclusion may be revised to focus on the role of P2X7R and describe how P2X7R lowers the risk of colitis-associated colorectal cancer.

R: We amended the text (conclusion) regarding the conclusion for clarity, as requested by this reviewer.

The abbreviations such as P2X7R, CRC or CA-CRC need to be consistent in the manuscript.

R: In lines 122 and 555, we substituted P2X7R for P2X7-R. Regarding CRC and CA-CRC, we changed some parts of the text but retained the specific meanings.

Please re-check the manuscript, especially Ca2+ in lines 44 and 61, etc.

R: We substituted Ca2+  for Ca in line 44, and Ca++.for Ca2+ in line 61, as pointed out by this reviewer.

Reviewer 4 Report

Comments and Suggestions for Authors

The herein presented review is focused on the P2X7 receptor in chronic inflammation and carcinogenic changes in the intestine. 

Introduction: 

At line 40 - can you add some examples of autoimmune disorders, as you cited two sources? To keep the sentence targeted to specific diseases. 

Figure 1 - a very good schematic representation of the P2X7 receptor in the intestinal mucosa and all the molecular pathways that are modulated. 

Chapter 2. A bit too short, but organized and focused on inflammatory bowel disease and CRC. 

Chapter 3. Lines 116-118 - I would recommend you to change the way you highlight your previous findings, avoid using We fin, We demonstrated, refer to your research group. 

Figure 2.  Well represented inflammation and neoplastic transformation in the gut. However, you should add in the legend also the immune cells that you depict there. 

Chapter 4. Presents in detail the abnormal innate immunity. Is contains a lot of information and I recommend you to make a table where to summarize for each innate immune cell, how they become abnormal and which effect they have on IBD and CRC. 

Chapter 5. Well organized and the inflammasome and DAMPs in IBD and CRC are clearly presented. 

Chapter 6. The structure is fine and I don't think that this needs improvements. 

Chapter 7. This chapter is one of interest, the link between intestinal microbiota and P2X7R seem to be important for key biological functions in the intestine. I recommend that you make a figure to depict this information. It would be useful for the readers and for sure it may increase the level of understanding the information even for those that are not to familiar with this topic, and highlight the pro or anti inflammatory profile depending on the diversity of microbiota. Or summarize it in a table, for each bacteria sp and say which are involved in IBD and which in CRC and the mechanisms that they may modulate or control.   

Conclusion. I have noting to comment here. The conclusion is fine. 

Author Response

REVIEWER 4

The herein presented review is focused on the P2X7 receptor in chronic inflammation and carcinogenic changes in the intestine. 

Introduction:

At line 40 - can you add some examples of autoimmune disorders, as you cited two sources? To keep the sentence targeted to specific diseases. 

R: We agree with this comment and have included information as suggested and three additional references.

DAMPs are also regarded as pro-inflammatory mediators that can amplify the inflammatory response, fueling the development of pathological changes underlying several inflammatory and autoimmune disorders, such as psoriasis, rheumatoid arthritis, and multiple sclerosis [4,5,6,7,8,9].

References

  1. de Carvalho Braga G, Francisco GR, Bagatini MD. Current treatment of Psoriasis triggered by Cytokine Storm and future immunomodulation strategies. J Mol Med (Berl). 2024 Oct;102(10):1187-1198. doi: 10.1007/s00109-024-02481-1. Epub 2024 Aug 30. PMID: 39212718.
  2. Milicevic KD, Bataveljic DB, Bogdanovic Pristov JJ, Andjus PR, Nikolic LM. Astroglial Cell-to-Cell Interaction with Autoreactive Immune Cells in Experimental Autoimmune Encephalomyelitis Involves P2X7 Receptor, β3-Integrin, and Connexin-43. Cells. 2023 Jul 5;12(13):1786. doi: 10.3390/cells12131786. PMID: 37443820; PMCID: PMC10340259.
  3. Li M, Yang C, Wang Y, Song W, Jia L, Peng X, Zhao R. The Expression of P2X7 Receptor on Th1, Th17, and Regulatory T Cells in Patients with Systemic Lupus Erythematosus or Rheumatoid Arthritis and Its Correlations with Active Disease. J Immunol. 2020 Oct 1;205(7):1752-1762. doi: 10.4049/jimmunol.2000222. Epub 2020 Aug 31. PMID: 32868411.
  4. Liu, X.; Li, Y.; Huang, L.; Kuang, Y.; Wu, X.; Ma, X.; Zhao, B.; Lan, J. Unlocking the therapeutic potential of P2X7 receptor: a comprehensive review of its role in neurodegenerative disorders. Front Pharmacol 2024, 15, 1450704, doi:10.3389/fphar.2024.1450704.

Figure 1 - a very good schematic representation of the P2X7 receptor in the intestinal mucosa and all the molecular pathways that are modulated.

R: We thank this reviewer for the comment.

Chapter 2. A bit too short, but organized and focused on inflammatory bowel disease and CRC.

R: We agree that Chapter 2 is relatively short. However, we did not mean to offer a comprehensive review of IBD and CRC specifically, as purinergic signaling was the central subject in the special issue.

Chapter 3. Lines 116-118 - I would recommend you to change the way you highlight your previous findings, avoid using We fin, We demonstrated, refer to your research group.

R: We understand this reviewer’s concern and amended the manuscript to comply with the suggestions.

Figure 2. Well represented inflammation and neoplastic transformation in the gut. However, you should add in the legend also the immune cells that you depict there.

R: We understand this reviewer’s concern, so we included additional labels in the figure and amended the legend accordingly for clarity.

Chapter 4. Presents in detail the abnormal innate immunity. Is contains a lot of information and I recommend you to make a table where to summarize for each innate immune cell, how they become abnormal and which effect they have on IBD and CRC.

R: As suggested, we included a table in Chapter 4 indicating the immune cells and their roles in IBD and CRC regarding the P2X7R modulation.

Chapter 5. Well organized and the inflammasome and DAMPs in IBD and CRC are clearly presented.

R: We thank this reviewer for the comment.

Chapter 6. The structure is fine and I don't think that this needs improvements.

R: We thank this reviewer for the comment.

Chapter 7. This chapter is one of interest, the link between intestinal microbiota and P2X7R seem to be important for key biological functions in the intestine. I recommend that you make a figure to depict this information. It would be useful for the readers and for sure it may increase the level of understanding the information even for those that are not to familiar with this topic, and highlight the pro or anti-inflammatory profile depending on the diversity of microbiota. Or summarize it in a table, for each bacteria sp and say which are involved in IBD and which in CRC and the mechanisms that they may modulate or control.

R: We understand this reviewer’s point of view and agree that a table or figure would, in theory, help the general understanding of the subject. However, considering that the number and diversity of participating microorganisms in such a context are so elevated, depicting interactions would not be a simple task. Therefore, we believe that such a task is beyond the scope of this manuscript and would probably constitute an exciting challenge to perform in a different setting and journal issue, with enough time to prepare. 

Conclusion. I have nothing to comment here. The conclusion is fine.

R: We thank this reviewer for the comment.